# CHAMELEON: ADAPTIVE CODE OPTIMIZATION FOR EXPEDITED DEEP NEURAL NETWORK COMPILATION

**Byung Hoon Ahn**[1]**, Prannoy Pilligundla**[1]**, Amir Yazdanbakhsh**[2]**, Hadi Esmaeilzadeh**[1]
[1] University of California, San Diego
[2] Google Research
`bhahn@eng.ucsd.edu, ppilligu@eng.ucsd.edu, ayazdan@google.com`
`hadi@eng.ucsd.edu`

## ABSTRACT

Achieving faster execution with shorter compilation time can foster further diversity and innovation in neural networks. However, the current paradigm of executing neural networks either relies on hand-optimized libraries, traditional compilation heuristics, or very recently genetic algorithms and other stochastic methods. These methods suffer from frequent costly hardware measurements rendering them not only too time consuming but also suboptimal. As such, we devise a solution that can learn to quickly adapt to a previously unseen design space for code optimization, both accelerating the search and improving the output performance. This solution dubbed **CHAMELEON** leverages reinforcement learning whose solution takes fewer steps to converge, and develops an adaptive sampling algorithm that not only focuses on the costly samples (real hardware measurements) on representative points but also uses a domain-knowledge inspired logic to improve the samples itself. Experimentation with real hardware shows that **CHAMELEON** provides 4.45×speed up in optimization time over AutoTVM, while also improving inference time of the modern deep networks by 5.6%.

## 1 INTRODUCTION

The enormous computational intensity of Deep Neural Networks (DNNs) have resulted in developing either hand-optimized kernels, such as NVIDIA cuDNN or Intel MKL that serve as backend for a variety of programming environment such as TensorFlow (Abadi et al., 2016) and PyTorch (Paszke et al., 2019). However, the complexity of the tensor operations in DNNs and the volatility of algorithms, which has led to unprecedented rate of innovation (LeCun, 2019), calls for developing automated compilation frameworks. To imitate or even surpass the success of hand-optimized libraries, recent research has developed stochastic optimization passes: for general code, STOKE (Schkufza et al., 2013), and neural network code, TVM (Chen et al., 2018a) and TensorComprehensions (Vasilache et al., 2018). TVM and TensorComprehensions are based on random or genetic algorithms to search the space of optimized code for neural networks. AutoTVM (Chen et al., 2018b) builds on top of TVM and leverage boosted trees (Chen & Guestrin, 2016) as part of the search cost model to avoid measuring the fitness of each solution (optimized candidate neural network code), and instead predict its fitness. However, even with these innovations the optimizing compilation time can be around 10 hours for ResNet-18 (He et al., 2016), and even more for deeper or wider networks.

Since the general objective is to unleash new possibilities by developing automatic optimization passes, long compilation time hinders innovation and could put the current solutions in a position of questionable utility. To solve this problem, we first question the very statistical guarantees which the aforementioned optimization passes rely on. The current approaches are oblivious to the patterns in the design space of schedules that are available for exploitation, and causes inefficient search or even converges to solutions that may even be suboptimal. Also, we notice that current approaches rely on greedy sampling that neglects the distribution of the candidate solutions (configurations). While greedy sampling that passively filter samples based on the fitness estimations from the cost models work, many of their hardware measurements (required for optimization) tend to be redundant and wasteful. Moreover, we found that current solutions that rely on greedy sampling lead to significant fractions of the candidate configurations being redundant over iterations, and that any optimizing

compiler are prone to invalid configurations which significantly prolongs the optimization time. As such, this work sets out to present an *Adaptive* approach dubbed **CHAMELEON** to significantly reduce the compilation time and offer automation while avoiding dependence to hand-optimization, enabling far more diverse tensor operations in the next generation DNNs. We tackle this challenge from two fronts with the following contributions:

(1) Devising an *Adaptive Exploration* module that utilizes reinforcement learning to adapt to unseen design space of new networks to reduce search time yet achieve better performance.
(2) Proposing an *Adaptive Sampling* algorithm that utilizes clustering to adaptively reduce the number of costly hardware measurements, and devising a domain-knowledge inspired *Sample Synthesis* to find configurations that would potentially yield better performance.

Real hardware experimentation with modern DNNs (AlexNet, VGG-16, and ResNet-18) on a high-end GPU (Titan Xp), shows that the combination of these two innovations, dubbed **CHAMELEON**, yields 4.45×speedup over the leading framework, AutoTVM. **CHAMELEON** is publicly available in the project page: `https://bitbucket.org/act-lab/chameleon`.

## 2 CHALLENGES IN DEEP NEURAL NETWORK COMPILATION

The general life-cycle of deep learning models from its birth to deployment comprises of two major stages. First stage is the designing and the training of a deep learning model by a research scientist, with the primary goal of achieving the highest feasible accuracy. Then, with a general demand to enable the intelligence on a wide range of devices (from mobile CPUs in the edge to cloud-scale GPUs), the second stage has emerged for the deployment of the pre-trained deep learning model to a target hardware by a deployment engineer. These stages are each iterative processes: research scientists iterate until it reaches the target performance in terms of accuracy whereas the deployment engineers iterate until the performance in terms of inference speed with a given hardware satisfies the given constraints. Importantly, these two stages are most often separate processes, and this paper mainly focuses on the second stage (deployment) of the cycle with an overarching goal of accelerating the overall deployment cycle by reducing the optimizing compilation time without compromising the performance of the output code.

### 2.1 COMPILATION WORKFLOW FOR DEEP NEURAL NETWORKS

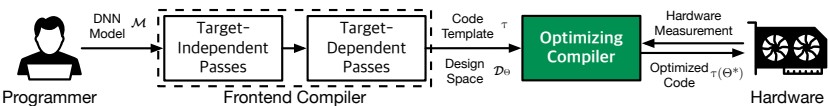

Figure 1: Overview of our model compilation workflow, and highlighted is the scope of this work.

Figure 1 illustrates how a compiler for DNNs takes an input model $\mathcal{M}$ and emits an optimized code $\tau(\Theta^*)$ that runs the model efficiently on a given hardware. This flow is commensurate with Tensor-Comprehensions (Vasilache et al., 2018) and TVM (Chen et al., 2018a), using which we implement **CHAMELEON** that is available as a separate package for adoption in even other frameworks. The first phase of the workflow is the frontend compiler which performs the translation from the compiler and applies target-independent and white-box target-dependent optimizations that do not incorporate a measure of runtime. Target-independent passes transform the input DNN model without specificity to the target hardware. Operator fusion and data layout transformation in TVM are some examples of these passes, which lie in the same category as dead-code elimination or loop-invariant code motion in GCC (Stallman & DeveloperCommunity, 2009) or LLVM (Lattner & Adve, 2004). Target-dependent passes, on the other hand, the compiler takes the hardware architecture (target) into account while optimizing the program; however, this also does not actively leverage runtime measures. The last stage is a black-box optimization pass, called *optimizing compiler*, that given a measure of performance at runtime from the hardware can further optimize the code. **CHAMELEON** falls in this class by offering an optimizing compiler that *adapts* to different design space to be more swift in optimizing deep neural networks compared to conventional approaches.

| KNOBS | DEFINITION |
|-------|-----------|
| tile_f, tile_y, tile_x | Factors for tiling and binding # of filters height, and width of feature maps. |
| tile_rc, tile_ry, tile_rx | Factors for tiling reduction axis such as # of channels, height, and width of filters. |
| auto_unroll_max_step | Threshold of number of steps in the loop to be automatically unrolled. |
| unroll_explicit | Explicitly unroll loop, this may let code generator to generate pragma unroll hint. |

Table 1: Knobs in the design space to optimize convolution.

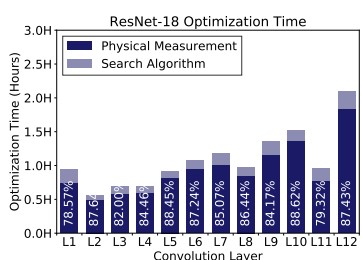

Figure 2: AutoTVM optimization time for ResNet-18 on Titan Xp.

## 2.2 OPTIMIZING COMPILER FOR DEEP NEURAL NETWORKS

Optimizing compilers (Kennedy & Allen, 2001) usually take a black-box approach and use hardware measurements to configure the optimization based on a measure of fitness $f$ of each solution. In order to make the problem tractable, the optimizing compilers for deep neural networks reduce the problem down to tuning the knobs $\theta$ for the output code template $\tau$, and can be formulated as:

$$\Theta^* = \underset{\Theta}{\operatorname{argmax}} f(\tau(\Theta)), \qquad \text{for } \Theta \in \mathcal{D}_\Theta. \tag{1}$$

A combination of assignment to the knobs is said to be a configuration $\Theta = (\theta_1, \theta_2, ..., \theta_n)$ while the dimensions of the design space $\mathcal{D}_\Theta$ is defined by the knobs. As such, in Equation 1, an optimizing compiler starts from a code template $\tau$ for each layer, and makes use of a search algorithm and real hardware measurements to efficiently find the best configuration $\Theta^* \in \mathcal{D}_\Theta$. In this context, there are three variables that determine the effectiveness of the optimizing compiler: (1) *a large and diverse enough design space that covers a variety of transformations*, (2) *an effective search algorithm to adequately navigate this space*, and (3) *a mechanism to cut down the number of costly hardware measurements that check the fitness of a solution*. Table 1 lists the knobs for performing convolution on a GPU, where it is crucial that the code (1) maximizes data reuse, (2) uses the shared memory wisely, and (3) minimizes bank conflicts. The knobs optimize various aspects of the execution, including tiling (e.g., tile_x, tile_y, ...), unrolling (e.g., auto_unroll_max_step and unroll_explicit), and these knobs define a design space with $10^{10}$ possibilities. Given the vastness of the design space, the remaining challenges are designing *an effective search algorithm* and designing *a mechanism that reduces the cost of each step in the search* (i.e. reducing the need to measure the hardware).

## 2.3 CHALLENGES IN DEEP NEURAL NETWORK COMPILATION

As shown in Figure 2, optimizing compilation for DNNs may still take an eon even with the advances from prior works (Chen et al., 2018a;b; Vasilache et al., 2018) With active research (You et al., 2017; Goyal et al., 2017; Codreanu et al., 2017; Akiba et al., 2017; You et al., 2018; Mattson et al., 2019) that has been able to cut down the training time to only few hours (You et al., 2017; Goyal et al., 2017) and even minutes (You et al., 2018; Akiba et al., 2017) on big models (e.g., ResNet-50 (He et al., 2016)) for ImageNet, it renders the optimizing compilation time of the current solutions seem even more prominent. Especially, since the above-mentioned compilers have been integrated to the deep learning pipelines of major players in the industry (Liu et al., 2019; Rotem et al., 2018; Vasilache et al., 2018), many users of these pipelines including the deployment engineers must go through the compilation workflow depicted in Figure 1 numerous times. Therefore, current long compilation time can be a hindrance to deploying DNN in various hardware, hence a major bottleneck in enabling intelligence on wider range of target platforms.

Furthermore, as we explore various neural topologies (Xie et al., 2019; Wortsman et al., 2019) for better performance as illustrated in Ahn et al. (2020), even deeper or wider networks (Szegedy et al., 2015; Zagoruyko & Komodakis, 2016), and new operations (Howard et al., 2017) to achieve higher performance (LeCun, 2019), we are forced to optimize the networks more frequently. The long optimization times are multiplied with such trend, leaving the practical utility of the current compiler solutions to question. As such, the primary goal of this work is reducing the optimizing compilation time to meet the immediate needs of the industry for expedited DNN compilation to foster further diversity and innovation in designing DNNs.

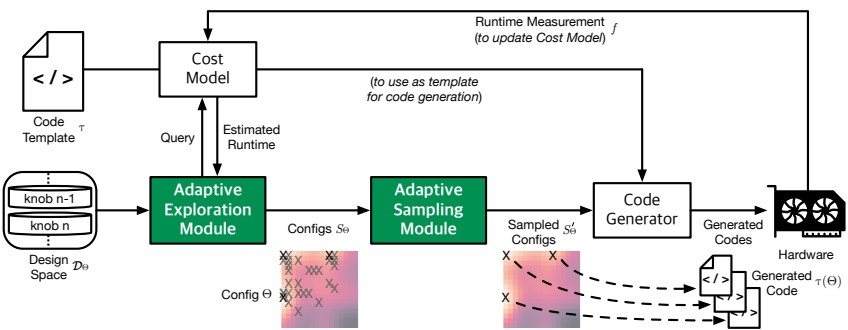

Figure 3: Overall design and compilation overview of the **CHAMELEON**.

Such long optimization time results from the inefficiency of simulated annealing which (while it stochastically guarantees a reasonable solution after huge number of iterations) *fails to capture the patterns in the design space* that can be exploited during the search. On the other hand, we can see in the figure that *majority of the optimization time is spent on reaching for measurements on real hardware* that is used as a feedback for the aforementioned search. Also, current approach even *suffers from numerous invalid configurations* that not only wastes the limited hardware measurement budget that the compiler starts with, but also incurs serious overhead to reset the target hardware for subsequent hardware measurements. As such, it is important that a sampling mechanism that selects potential configurations for hardware measurements to be smarter to ensure that each measurement is maximizing the chances of achieving a good solution and that it evades the invalid configurations. However, the current approaches rely on greedy sampling that passively sample based on the estimations from the cost models. This not only has a tendency to overfit but also neglect that solutions are distributed non-uniformly and that there are numerous invalid configurations.

# 3 CHAMELEON: ADAPTIVE CODE OPTIMIZATION FOR EXPEDITED DEEP NEURAL NETWORK COMPILATION

As discussed in Section 2, current solutions fall short of providing a swift optimization framework for optimizing emergent deep neural networks, because of the futility of the search in adapting to the design space from a random walk based search algorithm and the inefficiency of the physical hardware measurements from the greedy sampling. Therefore, developing a new framework that can overcome current challenges to unfetter neural network innovation from a prolonged optimization times can be boiled down to two problems: ❶ improving the the search algorithm to better adapt to the design space, and ❷ improving the sampling algorithm to both better adapt to the distribution of the solutions and decrease the possibility of running into invalid configurations. As such we make two innovations in the optimizing compiler for deep neural networks to develop **CHAMELEON** by applying reinforcement learning to the search that can adapt to new design spaces (*Adaptive Exploration*) and devising an *Adaptive Sampling* that replaces the current greedy sampling.

## 3.1 OVERALL DESIGN OF CHAMELEON

Figure 3 outlines the overall design of our optimizing compiler, dubbed **CHAMELEON**[1], and gives an overview of the optimizing compilation process. **CHAMELEON** takes code template $\tau$ for each layer in the network and the corresponding design space $\mathcal{D}_\Theta$ as its input, and iteratively optimizes the code for configuration $\Theta$ to finally output $\tau(\Theta^*)$. The proposed *Adaptive Exploration* maneuvers the design space while using a cost model as a proxy for hardware measurements to the output set of candidate configurations $S_\Theta$. These configurations are then sampled with *Adaptive Sampling* so that the sampled configurations $S'_\Theta$ subsume the initial candidate configurations while reducing its number significantly. The sampled configurations $S'_\Theta$ are then passed to the code generator which combines the input template $\tau$ and the configurations $S'_\Theta$ to create a set of $\tau(\Theta)$ that are sent to real hardware for runtime measurements. Runtimes from the hardware are used as the measure of fitness

---

[1]Chameleon is an animal that is capable of *Adapting* to their environments which helps them survive. In our work, **CHAMELEON** is an entity that *Adapts* to the variations in the design space and the distribution of the candidate configurations, enabling expedited deep neural network compilation.

$f$ and update the cost model to enhance the exploration of the subsequent iterations. After multiple iterations, $\tau(\Theta^*)$ with the best fitness $f$ (shortest runtime) is selected as an output for the layer.

## 3.2 Adaptive Exploration: Learning about the Unseen Design Space to Expedite Convergence of Optimization

As stated in Section 2, the current state-of-the-art approach (Chen et al., 2018b) that leverages simulated annealing relies on the stochastic guarantes of its random walks. Therefore, the current approach requires numerous iterations of exploration to converge to a reasonable solution causing long compilation hours, thus insufficient to enable disruptive innovations in neural networks. We take an inspiring approach that avoids naive dependence on the stochastic guarantee of simulated annealing and leverage a technique that can *learn to adapt* to unseen design space to not only accelerate convergence but also bring some performance gains. As such, we develop *Adaptive Exploration* by leveraging *Reinforcement Learning (RL)*, which is concerned with learning to maximize reward given an environment by making good *exploration* and *exploitation* tradeoffs, in our case maximizing fitness $f$ of the explored configurations $S_\Theta$.

**Reinforcement learning formulation.** Our RL-based Adaptive Exploration module uses an *actor-critic style RL*, where policy network learns to emit a set of directions (vector of increment/decrement/stay) for each knob in the design space that will increase $f$ of the next configuration and the value network learns the design space $\mathcal{D}_\Theta$ to estimate the value of the action. The first layer of these networks that takes the current configuration $\Theta$ as input is shared to foster information sharing among the two networks, and its output is fed into the subsequent layers the networks. These networks not only learn the dependencies among the different knobs of the design space (which are interrelated) that helps our module navigate through the design space but also lean the potential gains of the modifications to the configurations.

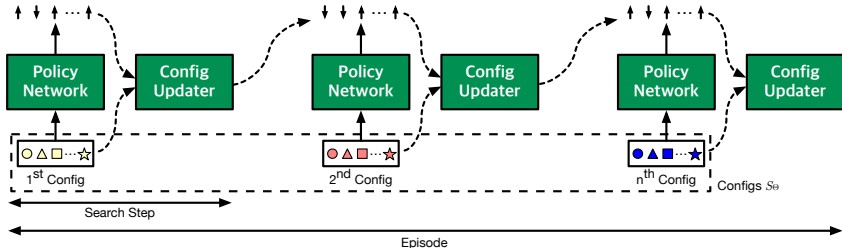

Figure 4: Adaptive Exploration Module of **Chameleon** in action.

**Learning procedure.** Having formulated the RL-based Adaptive Exploration Module, an iteration of our optimization begins with a set of initial configurations and takes multiple search steps (episode) for each of the configurations. As shown in Figure 4, the agent makes an action and applies it to the configuration using configuration updater to get another configuration that potentially has better $f$. After finishing multiple search steps in the episode, all configurations $S_\Theta$ are evaluated using a cost model, which its return values are used as a surrogate reward to update our agent, to reduce the number of costly hardware measurements. By taking this approach, $f$ of $S_\Theta$ improves as our module progresses through the episodes. In other words, by repeating multiple episodes and iterations, our Adaptive Exploration Module gradually learns to locate good configurations.

## 3.3 Adaptive Sampling: Adapting to the Distribution to Reduce Costly Hardware Measurements

**Reducing number of costly hardware measurements.** After the exploration step (regardless of the exploration method), we observe that the candidate configurations are clustered in subregions of the design space and these clusters are non-uniformly distributed (Figure 5). We also find that, while the design space's surface is discrete and un-smooth, a large fraction of configurations within each cluster achieve similar runtime (Figure 6). Utilizing these characteristics of the design space, we devise *Adaptive Sampling* that can sample a new set of candidates, by adapting to the shape of the

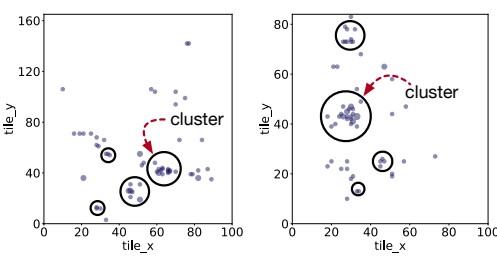

(a) VGG-16 4th layer     (b) ResNet-18 11th layer

Figure 5: Clusters of candidate configurations.

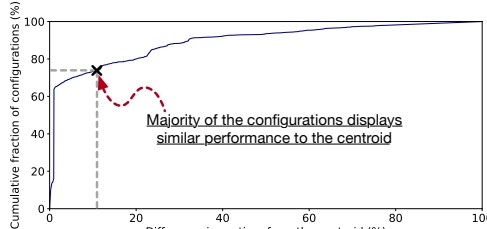

Figure 6: Cumulative Distribution Function (CDF) of the difference in runtime among the configurations in the cluster.

design space and the non-uniformity of the distribution while leaving the performance of optimization intact. We first leverage *clustering* algorithm to find configurations that are representative of each cluster; the sampling module uses centroids as the representative configurations. Our Adaptive Sampling iterates over a different number of clusters for their respective centroids and the L2 loss.

In the context of optimizing compiler, selecting the number of centroids for clustering entails making the important tradeoff between selecting more centroids for better performance or fewer centroids for a reduced number of hardware measurements. As such, we must devise a method that would automatically make the tradeoff in a reasonable manner. We take advantage of the decreasing trend in the aforementioned L2 loss as we increase the number of centroids, and devise a *Threshold-based Swift Meta-Search* to determine the number of clusters. By setting the threshold (hyperparameter) it allows the compiler to determine the point of diminishing return (*knee* of the curve), inflection point beyond which fewer centroids may lead to performance degradation and more clusters would prolong the optimization substantially. Overall, our sampling curtails the number of hardware measurements so that it is just enough to subsume the entire subspace of the candidate configurations.

**Improving candidate configurations using sample synthesis.** While the above sampling algorithm significantly reduces the number of hardware measurements compared to the conventional greedy sampling, without impacting the performance of the output code, we are still left with a critical issue of *redundancy among the candidate configurations*. We find that the exploration algorithm (regardless of the type) combined with the greedy sampling frequently leads to redundancy among the candidate configurations over different iterations of optimization due to the overfitting of the cost model from the greediness of the sampling. Even though the exploration algorithm tries to explore unvisited regions of the design space, these explored (not exploited) configurations are discarded due to the greedy sampling which entirely depends on the cost model for its selections of the configurations. Therefore, the current greedy sampling algorithm has its limitation in focusing the hardware measurements to the same region over and over.

On the other hand, we find that from a code optimization point of view, we know that many of the automated approaches for black-box optimization are prone to *invalid configurations*, which results from too large a tile that goes over the input feature map boundary or errors during memory accesses (cannot be solved analytically). These invalid configurations not only blow the chances for better exploration but also leads to an extra optimization time overhead to reset the physical hardware for the subsequent hardware measurement. We try to overcome both of these limitations by devising *Sample Synthesis*. When our compiler runs into redundant samples, the proposed synthesis method analyzes the candidate samples to determine the most probable (most frequent = mode function) non-invalid choice for each knob to come up with a new configuration. This statistical combination of the most frequent knob settings yield configurations that combine the strengths of different knobs to converge to a better overall solution. In spirit, the recombination (crossover) operator in genetic algorithms also tries to combine the best features of the solutions with high fitness values. Algorithm 1 presents the integration of our Adaptive Sampling and the Sample Synthesis.

### 3.4 IMPLEMENTATION DETAILS

**Architecture exploration for the adaptive exploration.** We use *Proximal Policy Optimization (PPO)* (Schulman et al., 2017), a policy gradient that has been shown to adapt to various problems and have good sample complexity, as our reinforcement learning algorithm. Since reinforcement

---

**Algorithm 1** Adaptive Sampling and Sample Synthesis

---

1: **procedure** ADAPTIVESAMPLING($s_\Theta, v_\Theta$)          ▷ $s_\Theta$: candidate configs, $v_\Theta$: visited configs
2:     new_candidates ← ∅, previous_loss ← ∞
3:     **for** $k$ **in** range(8, 64) **do**
4:         new_candidates, clusters, L2_loss ← K-means.run($s_\Theta, k$)
5:             **if** Threshold × L2_loss ≥ previous_loss **then** break        ▷ Exit loop at *knee* of loss curve
6:             previous_loss ← L2_loss
7:     **end for**
8:     **for** candidate **in** new_candidates **do**                    ▷ Replace visited config with new config
9:         **if** candidate **in** $v_\Theta$ **then** new_candidates.replace(candidate, mode($s_\Theta$))
10:     **end for**
11:     **return** new_candidates       ▷ Feed to *Code Generator* to make measurements on hardware
12: **end procedure**

---

learning could incur computational overhead that could prolong the optimization time, we optimize the actor-critic networks through architecture exploration to find good tradeoff for size of these networks (that determines the computational overhead) and the optimization performance.

**Design choices for the adaptive sampling.**    We use a $\mathcal{K}$-*means Clustering* to determine centroids of the configurations, because $\mathcal{K}$-means has been shown effective in practice and it only requires $\mathcal{K}$, over error $\epsilon$ or radius in other algorithms which are much more challenging to tune. For example, DBSCAN (Ester et al., 1996) or mean-shift clustering (Comaniciu & Meer, 2002) are very sensitive to the above hyperparameters. On the other hand, $\mathcal{K}$ can be framed as a *lever* to balance the performance and speed of optimizing compilation which abstracts away the aforementioned challenges, enabling the Threshold-based Swift Meta-Search that identifies the optimal number of clusters.

**Hyperparameter tuning.**    Hyperparameter tuning is a very important task in machine learning-based tools and models. As such, we present the hyperparameters we used for the evaluation in Table 7 (in appendix), which its tuning took several days. For the hyperparameters in Table 8 (in appendix), we used the same set of values that were used in the AutoTVM paper (Chen et al., 2018b) in order to conduct a fair comparison or **CHAMELEON**. Additionally, for parameters used in the Adaptive Exploration module, which is not present in AutoTVM, we have tuned the hyperparameters using the set of layers presented in Table 5 (in appendix). We emphasize, however, that the *hyperparameters have been tuned offline before the deployment of* **CHAMELEON**, and the hyperparameters are not changed during the use of the framework or the experimentation. So the tuning overhead is not part of the compilation after the Adaptive Exploration module is tuned once before releasing the compiler to the deployment practitioners.

## 4    EVALUATION

We integrate **CHAMELEON** into TVM (Chen et al., 2018a) to perform component evaluation and compare with AutoTVM (Chen et al., 2018b). We first evaluate components of **CHAMELEON** in Section 4.1 and Section 4.2 on set of convolution layers sampled from AlexNet (Krizhevsky et al., 2012), VGG-16 (Simonyan & Zisserman, 2015), and ResNet-18 (He et al., 2016). Then we provide end-to-end evaluation of **CHAMELEON** on both set of layers and end-to-end deep models, in Section 4.3. Due to space limitations, we present only the representative plots in the paper, and the complete set of results and the details of the parameters are provided in the appendix.

### 4.1    ADAPTIVE EXPLORATION: IMPROVING EFFICACY OF SEARCH ALGORITHM

In the previous approach (Chen et al., 2018b), authors have built a cost model to estimate fitness instead of performing costly measurements on real hardware, then used simulated annealing to find potentially optimal configurations. Figure 7(a) compares the number of search steps taken per iteration to reach or converge to the solution in simulated annealing and Adaptive Exploration, respectively. Overall, observation is that **CHAMELEON**'s Adaptive Exploration requires 2.88×less search steps compared to simulated annealing to find good solution. This comes from the ability of the re-

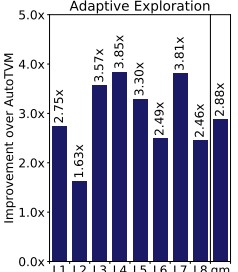
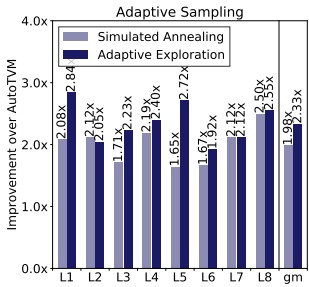
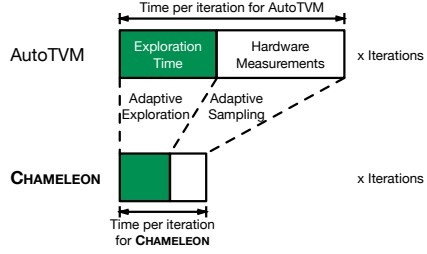

(a) Reduction in number of steps for convergence.

(b) Reduction in number of hardware measurements.

(c) Illustration of how the each component of **CHAMELEON** reduces the optimization time.

Figure 7: Component evaluation of **CHAMELEON**.

inforcement learning algorithm in Adaptive Exploration Module to (1) learn the correlation between different dimensions, and (2) reuse information across different iterations, instead of starting from scratch while naively relying on the stochastic guarantees of simulated annealing process.

## 4.2 Adaptive Sampling: Reducing Number of Costly Hardware Measurements

Figure 7(b) summarizes the effect of applying **CHAMELEON**'s Adaptive Sampling module on simulated annealing and reinforcement learning based search. First, the results show that using Adaptive Sampling helps the framework to make less hardware measurements regardless of the search algorithm used. The Adaptive Sampling algorithm reduces the number of measurements by $1.98\times$ when used with simulated annealing and $2.33\times$ with reinforcement learning One observation is that the Adaptive Sampling is more effective with reinforcement learning search. This comes from the reinforcement learning agent's capacity to better localize the search to meaningful samples (*exploitation*) while still aiming to find good solution by making diverse search (*exploration*).

Diversity exploration of AutoTVM aims to spread out the candidate configurations with a regularizing effect that fosters *uniform sampling*. In contrast, our Adaptive Sampling uses a clustering algorithm to perform more measurements on the regions with higher likelihood of achieving better output performance, leading to a *non-uniform sampling*. While AutoTVM states that diversity-aware selection had no meaningful impact on

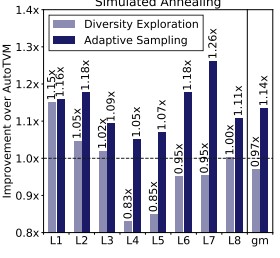
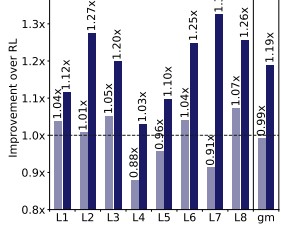

(a) Simulated Annealing.

(b) Reinforcement Learning.

Figure 8: Comparison to AutoTVM's diversity exploration.

most of the evaluated workloads, our Adaptive Sampling brings significant improvement as depicted in Figure 8. As shown, Adaptive Sampling brings an average of 13.5% and 19.0% improvement on simulated annealing and reinforcement learning, respectively.

## 4.3 Integration: Reducing Optimization Time and Output Inference Time

**CHAMELEON** integrates two components into the workflow: RL-based Adaptive Exploration (AE) and Adaptive Sampling (AS). This section compares the performance of **CHAMELEON** with AutoTVM (Chen et al., 2018b) that leverages Simulated Annealing (SA) for its exploration.

**Layer evaluation.** Figure 9 shows the trend of output code performance of ResNet-18's 11th layer over number of hardware measurements during optimization. The figure illustrates that our Adaptive Exploration finds better configurations than simulated annealing which results in better output code performance, and the Adaptive Sampling reduces number of hardware measurements significantly during optimization. Also, **CHAMELEON**'s Adaptive Exploration and Adaptive Sampling working in tandem emits better code with shorter optimization time than others. As such, Figure 10(a) compares optimization time and the performance of the output code in **CHAMELEON** and AutoTVM to confirm

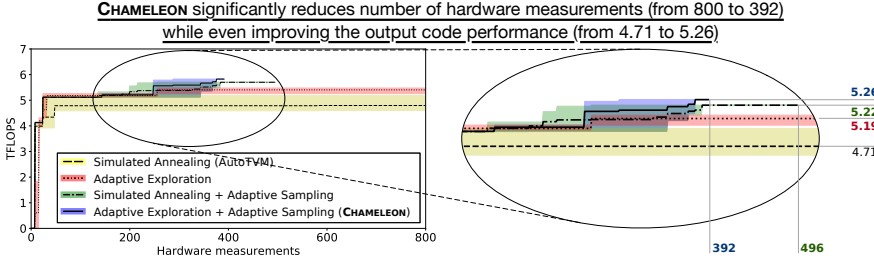

Figure 9: Layer evaluation of output performance for ResNet-18's 11th layer.

the observation. **CHAMELEON** achieved 1.17×better performance with 4.82×shorter optimization time compared to AutoTVM. Overall, the results suggest that our Adaptive Exploration effectively maneuvers the design space, and *Adaptive Sampling* reduces hardware measurements and the overall optimization time while even improving output performance.

**End-to-end evaluation.** Up until now, we have focused on evaluation with subset of layers. Now we continue our discussion to the applicability of **CHAMELEON** to optimization of end-to-end deep neural networks. Figure 10(b) shows that **CHAMELEON** spends 3.59×, 5.73×, and 4.28×less time than AutoTVM to optimize AlexNet, VGG-16, and ResNet-18, respectively. On average, our work shows 4.45×optimization time speedup while achieving up to 6.4% improvement in terms of performance of output code. Inference time in Figure 10(b) illustrates the speedup for optimized code. Raw numbers are available in Table 2 and Table 3. All in all, such improvements result from efficient Adaptive Exploration and the reduced number of hardware measurements from Adaptive Sampling.

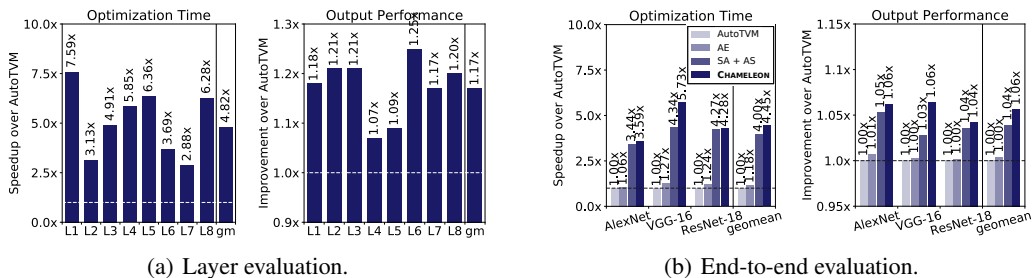

(a) Layer evaluation.        (b) End-to-end evaluation.

Figure 10: Layer and end-to-end evaluation. Dashed lines denote AutoTVM's performance.

| NETWORK | SA (AutoTVM) | AE | SA + AS | AE + AS (CHAMELEON) |
|---|---|---|---|---|
| AlexNet | 4.31 Hours | 4.06 Hours | 1.25 Hours | **1.20 Hours** |
| VGG-16 | 11.18 Hours | 8.82 Hours | 2.57 Hours | **1.95 Hours** |
| ResNet-18 | 9.13 Hours | 7.39 Hours | 2.14 Hours | **2.13 Hours** |

Table 2: End-to-end evaluation of the optimization time for deep networks.

| NETWORK | SA (AutoTVM) | AE | SA + AS | AE + AS (CHAMELEON) |
|---|---|---|---|---|
| AlexNet | 1.0277 ms | 1.0207 ms | 0.9762 ms | **0.9673 ms** |
| VGG-16 | 3.9829 ms | 3.9710 ms | 3.8733 ms | **3.8458 ms** |
| ResNet-18 | 1.0258 ms | 0.9897 ms | 0.9897 ms | **0.9831 ms** |

Table 3: End-to-end evaluation of the output performance for deep networks.

# 5 RELATED WORKS

**CHAMELEON** uniquely offers a solution that exclusively enables (i) *Reinforcement Learning* and (ii) *Sampling* in the context of (iii) *Optimizing Compilers* for neural networks. As such, we discuss the related work from each of the three independent research directions.

**Optimizing compilers.** TensorComprehensions (Vasilache et al., 2018) and TVM (Chen et al., 2018a) use genetic algorithm and simulated annealing to choose parameters of polyhedral optimization for neural networks. In a more general context, some computing libraries (Whaley & Dongarra, 1998; Frigo & Johnson, 1998) make use of black box optimization and also profiling-based compilation passes (Chang et al., 1991; Novillo, 2014) utilize runtime information to generate optimized code. Later, AutoTVM (Chen et al., 2018b) incorporates learning with boosted trees within the cost model for TVM to reduce the number of real hardware measurements. While **CHAMELEON** is inspired and builds on these prior works, unlike them, it is based on reinforcement learning for *Adaptive Exploration*, and *Adaptive Sampling* that leverages clustering to reduce the number of measurements.

**Reinforcement learning for hyper-parameter optimization.** There are a growing body of studies on using reinforcement learning to perform various optimizations (Gao et al., 2018; Mirhoseini et al., 2017; Nareyek, 2003; Mao et al., 2016; Xu et al., 2018; Mao et al., 2019) for a variety of objectives including hyper-parameter optimization for neural networks. For instance, DeepArchitect (Negrinho & Gordon, 2017) and NAS (Zoph & Le, 2017) use reinforcement learning to automate the process of designing deep neural network models and their associated parameters. HAQ (Wang et al., 2019) and ReLeQ (Elthakeb et al., 2018) use reinforcement learning to chose levels of quantization for the layers of a given deep neural network. AMC (He et al., 2018) formulates neural network compression as a RL problem. A most recent effort (Paliwal et al., 2020)–which will be published concurrent to ours in ICLR 2020–combined RL with graph neural networks and genetic algorithms to optimize DNN execution. Our work exclusively explores a different problem, that is optimizing compilers using reinforcement learning.

**Sampling algorithms for learning.** Active learning is a broad field (Settles, 2009; Cohn et al., 1996; Sugiyama, 2006; Cai et al., 2013; Goetz et al., 2018; Wu et al., 2019) that uses a measure of the change in the model to decide which training data elements should be used to update the model. Passive learning (Yu & Kim, 2010; O'Neill et al., 2017) is an alternative view that independent of the model, analyze the distribution of the training data set and selects a subset. The Adaptive Sampling algorithm for **CHAMELEON** shares similarities with Passive learning but it differs in its context. The sampling is designed to reduce the number of samples (configuration) for hardware measurement from the exploration of the design space whilst performing an optimization to accelerate the process.

## 6 CONCLUSION

We present **CHAMELEON** to allow optimizing compilers to adapt to unseen design spaces of code schedules to reduce the optimization time. This paper is also an initial effort to bring *reinforcement learning* to the realm of optimizing compilers for neural networks, and we also develop an *Adaptive Sampling* with domain-knowledge inspired *Sample Synthesis* to not only reduce the number of samples required to navigate the design space but also augment its quality in terms of fitness. Experimentation with real-world deep models shows that **CHAMELEON** not only reduces the time for compilation significantly, but also improves the quality of the code. This encouraging result suggests a significant potential for various learning techniques to optimizing deep learning models.

## ACKNOWLEDGEMENT

We thank the anonymous reviewers for their insightful comments. We also thank Jinwon Lee and Jangho Kim for the fruitful discussions and feedbacks on the manuscript. This work was in part supported by generous gifts from Qualcomm, Google, Microsoft, and Xilinx as well as the Semiconductor Research Corporation (SRC) contract #2019-SD-2884, National Science Foundation (NSF) awards CNS#1703812, ECCS#1609823, CCF#1553192, Air Force Office of Scientific Research (AFOSR) Young Investigator Program (YIP) award #FA9550-17-1-0274, National Institute of Health (NIH) award #R01EB028350, and Air Force Research Laboratory (AFRL) and Defense Advanced Research Project Agency (DARPA) under agreement number #FA8650-20-2-7009. The U.S. Government is authorized to reproduce and distribute reprints for Governmental purposes notwithstanding any copyright notation thereon. The views and conclusions contained herein are those of the authors and should not be interpreted as necessarily representing the official policies or endorsements, either expressed or implied, of AFRL, DARPA or the U.S. Government.

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

APPENDIX

## A EXPERIMENTAL SETUP

### A.1 DNN MODELS AND LAYERS

Table 4: Details of the DNN models used in evaluating **CHAMELEON**.

| NETWORK | DATASET | NUMBER OF TASKS |
|---------|---------|-----------------|
| AlexNet | ImageNet | 5 |
| VGG-16 | ImageNet | 9 |
| ResNet-18 | ImageNet | 12 |

Table 5: Details of the layers used in evaluating **CHAMELEON**.

| NAME | MODEL | LAYER TYPE | TASK INDEX |
|------|-------|-----------|-----------|
| L1 | AlexNet | convolution | 1 |
| L2 | AlexNet | convolution | 4 |
| L3 | VGG-16 | convolution | 1 |
| L4 | VGG-16 | convolution | 2 |
| L5 | VGG-16 | convolution | 4 |
| L6 | ResNet-18 | convolution | 6 |
| L7 | ResNet-18 | convolution | 9 |
| L8 | ResNet-18 | convolution | 11 |

### A.2 HARDWARE SPECIFICATION

Table 6: Details of the hardware used for evaluation of **CHAMELEON**.

| SPECIFICATIONS | DETAILS |
|----------------|---------|
| GPU | Titan Xp |
| Host CPU | 3.4G Hz Intel Core i7 |
| Main Memory | 32GB 2400 MHz DDR3 |

## A.3 HYPER-PARAMETERS

Table 7: Hyper-parameters uses in **CHAMELEON**.

| HYPERPARAMETER | VALUE | DESCRIPTION |
|---|---|---|
| $iteration_{opt}$ | 16 | number of iterations for optimization process |
|  |  | (equivalent to 1000 hardware measurements) |
| $mode_{GBT}$ | xgb-reg | type of loss used for cost model |
| $b_{GBT}$ | 64 | maximum batch size of planning in GBT (Chen & Guestrin, 2016) |
|  |  | cost model per iteration of optimization process |
| $episode_{rl}$ | 128 | number of episodes for reinforcement learning |
| $step_{rl}$ | 500 | maximum steps of one reinforcement learning episode |
| $threshold_{meta}$ | 2.5 | threshold used for meta-search in sampling |

Table 8: Hyper-parameters uses in AutoTVM (Chen et al., 2018b).

| HYPERPARAMETER | VALUE | DESCRIPTION |
|---|---|---|
| $\Sigma(b_{GBT})$ | 1000 | total number of hardware measurements |
| $mode_{GBT}$ | xgb-reg | type of loss used for cost model |
| $b_{GBT}$ | 64 | batch size of planning in GBT (Chen & Guestrin, 2016) |
| $n_{sa}$ | 128 | number of Markov chains in parallel simulated annealing |
| $step_{sa}$ | 500 | maximum steps of one simulated annealing run |

Table 9: Hyper-parameters used in **CHAMELEON**'s PPO (Schulman et al., 2017) search agent.

| HYPERPARAMETER | VALUE |
|---|---|
| Adam Step Size | $1 \times 10^{-3}$ |
| Discount Factor | 0.9 |
| GAE Parameter | 0.99 |
| Number of Epochs | 3 |
| Clipping Parameter | 0.3 |
| Value Coefficient | 1.0 |
| Entropy Coefficient | 0.1 |

# B ADDITIONAL EXPERIMENTAL RESULTS

## B.1 OPTIMIZATION TIME BREAKDOWN FOR DNN MODELS

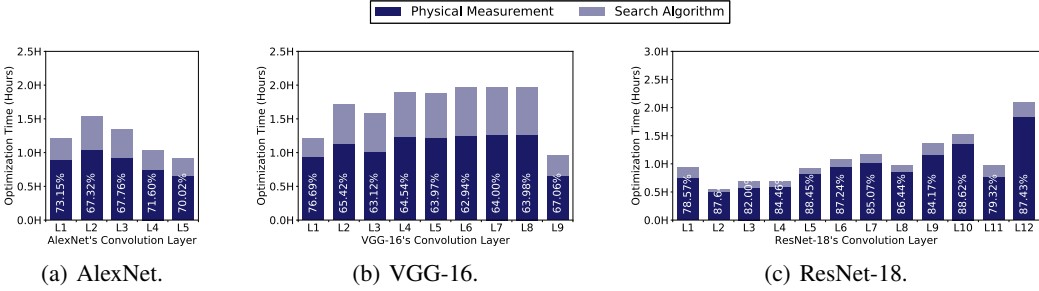

Figure 11: AutoTVM optimization time for AlexNet (Krizhevsky et al., 2012) and VGG-16 (Simonyan & Zisserman, 2015), and ResNet-18 (He et al., 2016) on Titan Xp. Numbers in bars denote fraction of time for measurements.

## B.2 PERFORMANCE VS. NUMBER OF MEASUREMENTS FOR DNN MODELS

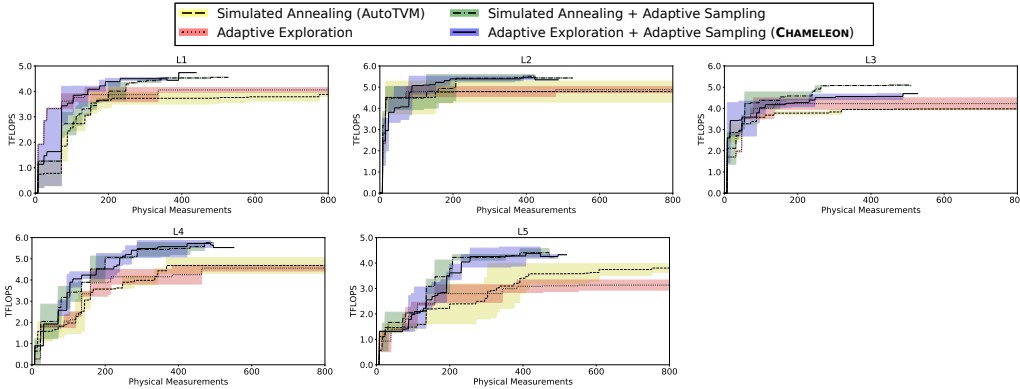

Figure 12: Layer evaluations for AlexNet (Krizhevsky et al., 2012).

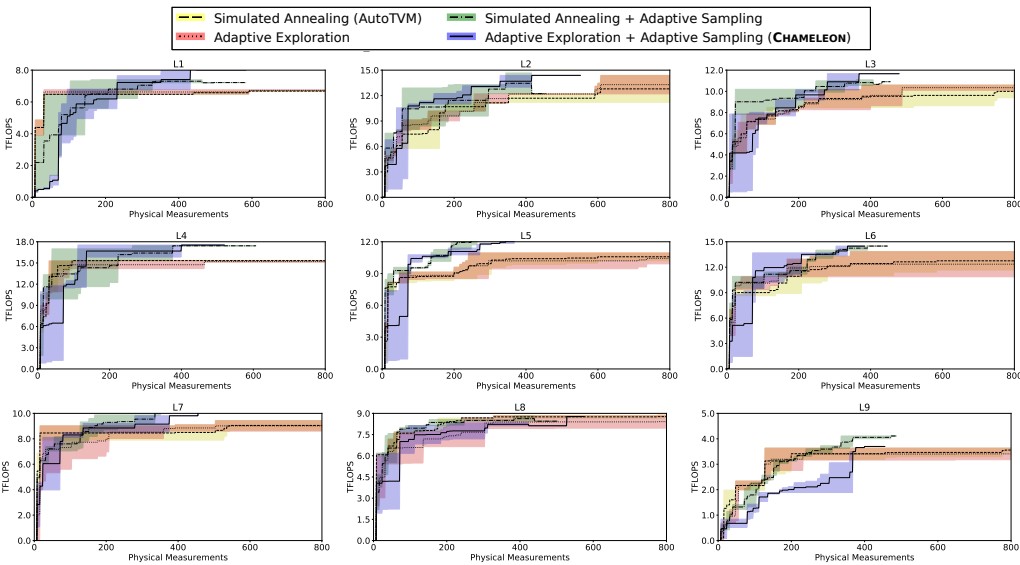

Figure 13: Layer evaluations for VGG-16 (Simonyan & Zisserman, 2015).

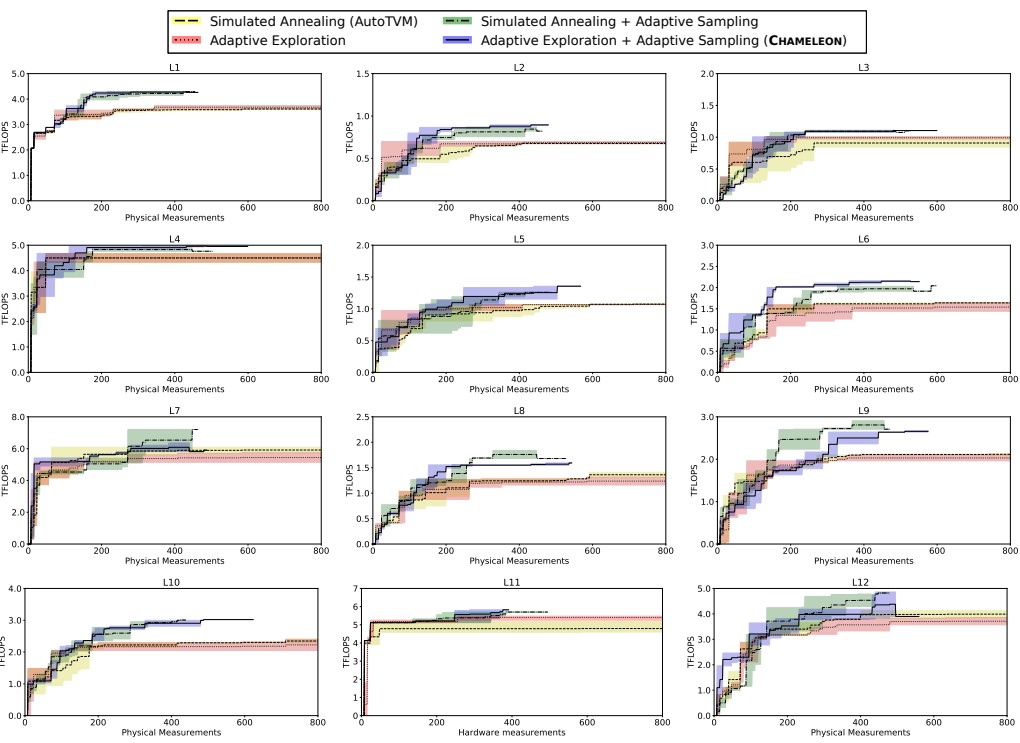

Figure 14: Layer evaluations for ResNet-18 (He et al., 2016).

