# OpenReview forum: "Chameleon: Adaptive Code Optimization for Expedited Deep Neural Network Compilation"
_ICLR.cc/2020/Conference — Accept (Poster)_

### Official Review · AnonReviewer3 · 2019-10-17
**Official Blind Review #3**

**Rating:** 6

**Review:**

The paper proposes a new solution called CHAMELEON for deep learning code optimization, which accelerates the process of compiling codes and achieves faster training and inference of deep networks. The proposed method can be used to compile various deep network architectures. Experimental results show that the proposed method outperforms the previous method with large margin.

The paper exploits reinforcement learning to address code compilation, which is novel for me.

The experimental results are convincing, the paper evaluates the proposed evaluation in multiple aspects.

I am totally new in the area. I will refer to the other reviews' review and the authors' rebuttal to have my final decision.


Post-rebuttal
Thank the other two reviewers and the authors to help me better understand the paper. I think I have no concern on the paper so I still give 6.

**Experience Assessment:**

I do not know much about this area.

**Review Assessment: Checking Correctness Of Derivations And Theory:**

I assessed the sensibility of the derivations and theory.

**Review Assessment: Checking Correctness Of Experiments:**

I assessed the sensibility of the experiments.

**Review Assessment: Thoroughness In Paper Reading:**

I read the paper at least twice and used my best judgement in assessing the paper.

---

> ### Author Response · Authors · 2019-11-11
> **Author Response**
>
> We thank the reviewer for a positive approach. We look forward to answering any questions or comments that may arise.

---

### Official Review · AnonReviewer1 · 2019-10-23
**Official Blind Review #1**

**Rating:** 6

**Review:**

The authors proposed a method for code optimization for deploying neural networks. The main idea is to formulate it as a search task over tuning knobs in the code template, and to apply reinforcement learning to optimize the configurations for the tuning knobs with respect to a cost model. The cost model is trained based on a subset of representative samples from the RL controller and their corresponding hardware cost measurements.

The paper is very well written and the proposed method seems technically sound. The authors did a good job combining existing techniques in a complementary manner. For example, the usage of RL for efficient search space exploration and the usage of clustering for selecting representative samples for on-device measurements.

Some concerns:
* The authors are referring to the usage of reinforcement learning as Adaptive Exploration and the usage of clustering as Adaptive Sampling. While combining them to tackle the task of neural network compilation is interesting, these techniques themselves are very standard and hence come with limited technical novelty.
* The proposed workflow seems to involve a nontrivial amount of additional hyperparameters, e.g., those in the RL controller as well as those for clustering. It might be useful to discuss about the overhead caused by hyperparameter tuning, as otherwise numbers reported in Table 2 (based on a single trial) could be misleading.


**Experience Assessment:**

I do not know much about this area.

**Review Assessment: Checking Correctness Of Derivations And Theory:**

I carefully checked the derivations and theory.

**Review Assessment: Checking Correctness Of Experiments:**

I assessed the sensibility of the experiments.

**Review Assessment: Thoroughness In Paper Reading:**

I read the paper at least twice and used my best judgement in assessing the paper.

---

> ### Author Response · Authors · 2019-11-11
> **Author Response**
>
> Thank you for insightful and encouraging comments.
>
> ===Contributions===
> This work is an initial step towards using Reinforcement Learning (RL) in an adaptive manner to optimize deep learning models. We agree with the reviewer that Reinforcement Learning (RL) nor clustering are new techniques. Nonetheless, their composition in a way that enables expedited DNN compilation as well as DNN execution is a new territory.
>
> In fact, the state-of-the-art Apache TVM compiler relies on Simulated Annealing (AutoTVM), which due to its random walks and high stochasticity does not adapt and incorporate to the underlying structure of the problem. Similarly, a straightforward basic application of RL with rather random walks (Basic RL) would not be beneficial as the included Table in this response illustrates. However, our definition of actions makes the exploration using RL faster by considering the domain-knowledge that many potential solutions (samples) are invalid and do not correspond to an executable code as follows: choosing a tile size or unrolling  factor can make the kernel out-of-range for a particular hardware. A Basic RL that permits the agent to take actions that can arbitrarily change any of the parameters (i.e. taking random walks), would lead the agent to many useless invalid configurations. As the Table shows, the Basic RL with the random walk for exploration is 19.2% slower in each iteration than AutoTVM even after using all of its strategies to filter out invalid configurations and even converges to underperforming solutions. CHAMELEON, in contrast, incorporates the domain knowledge in the composition of RL-based Adaptive Exploration and clustering-based Adaptive Sampling as follows.
>
> ———————————————————————————————
> Exploration algorithm               | Basic RL | AutoTVM | Our method
> ———————————————————————————————
> Time per iteration in seconds  |     204.0  |        171.2 |            138.6
> ———————————————————————————————
>
> (1: Increment-Based Action Space for Adaptive Exploration) The RL formulation in our Adaptive Exploration module only permit incremental changes to each template configurations when the agent acts and sidesteps a large faction of these invalid samples, leading to an RL formulation that even without sampling is slightly faster than AutoTVM
>
> (2: Non-uniform Adaptive Sampling) Building on this domain-specific adaptive formulation, we, exclusively, identify and leverage the insight that there is an opportunity to perform non-uniform sampling (through clustering) that leads to similar or better coverage with significantly fewer subsamples.
>
> (3: Domain-knowledge inspired Sample Synthesis) When it comes to compilation, the repeated nature of many samples (configuration) leads any exploration method combined with a greedy strategy to develop a greedy bias towards redundant regions. We alleviate this greedy bias through Sample Synthesis which analyzes and synthesizes a new sample that yields configurations that combine the strengths of different knobs to converge to a better overall solution (like recombination or crossover operator in genetic algorithms).
>
> (4: Applicability of Adaptive Sampling to other exploration strategies) In fact, the results shows that this non-uniform sampling is useful for optimized compilation of DNNs even with other conventional exploration techniques such as simulated annealing (used in AutoTVM). This added bonus is another evidence for our insight that the design space of DNN compilation lends itself to the Adaptive Sampling as developed in this paper.
>
> Overall, making RL work faster through (1) choosing an action space for Adaptive Exploration that considering the domain knowledge to sidestep invalid configurations (2) devising a non-uniform Adaptive Sampling and (3) leveraging the shape of the design space in DNN compilation and coming up with a domain-knowledge inspired Sample Synthesis that benefits from non-uniform Adaptive Sampling even when RL is not used, highlights the contributions.
>
> ===Hyperparameters===
> Reviewer is right that the CHAMELEON introduces additional hyperparameters, and we did indeed spend several days tuning the hyperparameters. This is an offline process and the hyperparameters are not changed during the use of the framework or the experimentation. So the tuning overhead is not part of the compilation after the Adaptive Exploration module is tuned once before releasing the compiler to the deployment practitioner (Revised in Section 3.4).
>
> The reported results in Table 2 and Table 3 are the average five independent trials as well as the results in Figures 12 through 14, in which the shades around the curves denote the confidence intervals.

---

### Official Review · AnonReviewer2 · 2019-10-23
**Official Blind Review #3**

**Rating:** 3

**Review:**

This paper proposes an optimizing compiler  for DNN's based on adaptive sampling and reinforcement learning, to drive the search of optimal code in order to reduce compilation time as well as potentially improve the efficiency of the code produced. In particular, the paper proposes to use PPO to optimize a code optimization "search" policy, and then use K-mean clustering over a set of different proposed compilation proposals, from which to perform adaptive sampling to reduce compilation time while still keeping a high diversity of the proposed solution pools during exploration. At the same time the authors claim that using RL will learn a better search strategy compared to random search - such as simulated annealing which is used by competing methods - thus producing faster and better solutions.
The paper show results of up to 4x speedup in compilation time (autotuning time) while obtaining a slightly better or similar efficiency of the generated code (in term of execution time). This is a well written extensive research with good results. The authors mention it is (will be) integrated in the open source code of TVM. However I could find no mention in the paper of whether the code will be released with this publication, and I would like to solicit the authors to clarify their code release strategy and timing.
My other question pertains to whether or not  compilation time is an key metric to target. It is important to some extent, but I would say that aside from exponential / super-polynomial behaviour of auto-tuning algorithms, a multiple hours / days process to create the best optimized code for a certain network / hardware platform might not be such a big hurdle for a community already used to multiple days / weeks / months to train the same models. I believe that focusing on the efficiency of the optimized code produced would probably be a better metric of success.

**Experience Assessment:**

I have read many papers in this area.

**Review Assessment: Checking Correctness Of Derivations And Theory:**

I assessed the sensibility of the derivations and theory.

**Review Assessment: Checking Correctness Of Experiments:**

I assessed the sensibility of the experiments.

**Review Assessment: Thoroughness In Paper Reading:**

I read the paper thoroughly.

---

> ### Author Response · Authors · 2019-11-11
> **Author Response**
>
> Thank you for the insightful and stimulating comments.
>
> ===Compilation Time and Training Time===
> (1) We agree with the reviewer that it used to take days and weeks to train a model, however with algorithmic advances [1] and hardware improvements, it is possible to train ResNet-50 in 29 hours with only 8 Tesla P100 GPUs [2]. According to MLPerf [3], the training time for ResNet-50 ranges between 2 hours (a single DGX-1) to 1.33 minutes (96 DGX-2H). Recently, fast.ai provides the means to “Training Imagenet in 3 hours for \$25; and CIFAR10 for \$0.26 [4].” That is, the trends are changing and, as the certain sub-areas mature, new challenges arise and spending 10 hours to just optimize the code for ResNet-18 (which has fewer layers than ResNet-50) becomes more prominent, hence not desirable anymore.
>
> —————————————————————————
> Paper                        |  Hardware                | Time
> —————————————————————————
> He et al. [2]              | Tesla P100 x 8          | 29 hours
> Goyal et al. [5]         | Tesla P100 x 256     |    1 hour
> Codreanu et al. [6] | KNL 7250 x 720        | 62 minutes
> You et al. [7]            | Xeon 8160 x 1600    | 31 minutes
> Akiba et al. [8]         | Tesla P100 x 1024   | 15 minutes
> —————————————————————————
>
> (2)  Furthermore, from direct academic and industry interactions, it is very common that the practitioners who design and train DNNs are not the ones who deploy them, which is usually on a variety of platforms and requires optimizing DNN execution according to hardware features and constraints. For example, (i) network designer A would design the model and saves them as a “pre-trained model” then (ii) deployment engineer B would work on porting the “pre-trained model” to various devices including smart phones, sensor systems, and other edge devices. In such scenario deployment is not a single shot process but an iterative process requiring possibly ~100 times of compilation based on first-hand industry experience.
>
> (3) Our work provides around 1.6x speedup over Tensorflow+cuDNN. In other words, both existing solutions and CHAMELEON achieve near optimal DNN execution time. However, our work tackles the next arising problem, which is the lengthy compilation time in state-of-the-art optimizing compilers while even squeezing the last drops of improvement in output code efficiency.
>
> (4) A prominent company has requested to integrate our work in the Apache TVM to address their compiler time issues. We understand that this is anecdotal evidence but it illustrates the importance of the problem domain and the need for research that addresses this emerging challenge: expedited compilation time for DNNs. We are more than happy to provide concrete evidence for the aforementioned industry request.
>
> We apologize for not clarifying these points. We have revised Section 2 and 2.3 to reflect upon the review.
>
> [1] You, Yang, Igor Gitman, and Boris Ginsburg. "Scaling sgd batch size to 32k for imagenet training." arXiv preprint arXiv:1708.03888 6 (2017).
> [2] He, Kaiming, et al. "Deep residual learning for image recognition." Proceedings of the IEEE conference on computer vision and pattern recognition. (2016).
> [3] Mattson, Peter, et al. "MLPerf Training Benchmark." arXiv preprint arXiv:1910.01500 (2019). https://mlperf.org
> [4] “Training Imagenet in 3 hours for \$25; and CIFAR10 for \$0.26”, fast.ai, https://www.fast.ai/2018/04/30/dawnbench-fastai/
> [5] Goyal, Priya, et al. "Accurate, large minibatch sgd: Training imagenet in 1 hour." arXiv preprint arXiv:1706.02677 (2017).
> [6] Codreanu, V., D. Podareanu, and V. Saletore. "Achieving deep learning Training in less than 40 minutes on ImageNet-1K & best accuracy and training time on ImageNet-22K & Places-365 with scale-out Intel® Xeon®/Xeon Phi™ architectures." (2017).https://blog.surf.nl/en/imagenet-1k-training-on-intel-xeon-phi-in-less-than-40-minutes/
> [7] You, Yang, et al. "Imagenet training in minutes." Proceedings of the 47th International Conference on Parallel Processing. ACM, (2018).
> [8] Akiba, Takuya, Shuji Suzuki, and Keisuke Fukuda. "Extremely large minibatch SGD: training resnet-50 on imagenet in 15 minutes." arXiv preprint arXiv:1711.04325 (2017).
>
> ===Code release===
> We have updated the manuscript to include a link to the anonymous repository (https://github.com/anony-sub/chameleon), and we not only will open source the code but also integrate it to the main branch of the Apache TVM at the request of a prominent and independent corporation and also to engage the community in this line of research.

---

### Author Response · Authors · 2019-11-15
**To Reviewers and the Area Chair**

We thank all the reviewers for their encouraging comments. We have addressed all the comments and feedback from the reviewers in our revision and provided a detailed answer in the comments section.

===(Reviewer 1) Open-Source Public Code Release===
We have updated the manuscript to include a link to the anonymous repository (https://github.com/anony-sub/chameleon), and we will not only open source the code but also integrate it to the main branch of the Apache TVM at the request of a prominent and independent company and also to engage the broader community in research and development in this emerging area. More details are provided in the inlined response.

===(Reviewer 1) Hyperparameters===
The hyperparameter tuning for CHAMELEON is an offline process and the hyperparameters are not changed during the use of the framework or the experimentation. So the tuning overhead (days) is not part of the compilation after the Adaptive Exploration module is tuned once before releasing the compiler to the deployment practitioner.

Furthermore, we clarify that the reported results in Table 2 and Table 3 are the average  over five independent trails as well as the results in Figures 12 through 14, in which the shades around the curves denote the confidence intervals. Please see more details in the inlined response.

===(Reviewer 2) Compilation Time and Training Time===
While it used to take days and weeks to train a model, with algorithmic advances and hardware development, training time of deep networks have reduced from months/weeks to hours/minutes [see the inlined response for the  detailed citations].  At  the same time, the deployment of DNNs have become a separate iterative task that requires numerous iterations (~100) of optimizing compilation for efficient deployment. The large compilation time (e.g., 10 hours for ResNet-18) is an emerging
challenge that is tackled in this paper.  As deep learning becomes more prevalent, this emerging problem, if not addressed, can curtail both innovation in deep learning and utility of those innovation on a wider range of applications and platforms.

Our solution CHAMELEON provides 4.45x speed up in optimization time while also improving inference time of the modern deep networks by 5.6% compared to AutoTVM and 60.4% over Tensorflow+cuDNN.

===(Reviewer 3)===
We thank the reviewer for the positive feedback and hope to answer any question that may come up.

---

### Decision · Program_Chairs · 2019-12-19

**Decision:**

Accept (Poster)

**Comment:**

This paper proposes to optimize the code optimal code in DNN compilers using adaptive sampling and reinforcement learning. This method achieves  significant speedup in compilation time and execution time. The authors made strong efforts in addressing the problems raised by the reviewers, and promised to make the code publicly available, which is of particular importance for works of this nature.